# Modal Parameter Identification and Comfort Assessment of GFRP Lightweight Footbridges in Relation to Human–Structure Interaction

Jordi Uyttersprot [1] , Wouter De Corte [1,*] and Wim Van Paepegem [2]

1  Department of Structural Engineering and Building Materials, Faculty of Engineering and Architecture, Ghent University, Tech Lane Ghent Science Park 60, 9052 Ghent, Belgium; jordi.uyttersprot@ugent.be
2  Department of Materials, Textiles and Chemical Engineering, Faculty of Engineering and Architecture, Ghent University, Tech Lane Ghent Science 46, 9052 Ghent, Belgium; wim.vanpaepegem@ugent.be
*  Correspondence: wouter.decorte@ugent.be

**Abstract:** With the emergence of slimmer footbridges and the introduction of lighter materials, the challenge of vibrational comfort assessment becomes more and more relevant. Previous studies have shown that each pedestrian will act both as an inducer and a damper, referred to as human–structure interaction. However, this interaction is currently not implemented in design guidelines, which leads to a poor comfort estimation for small lightweight footbridges. Derived from smartphone-based vibration measurements, this paper provides an overview of the modal parameters at various pedestrian densities and a comfort assessment of a selection of simply supported GFRP and steel lightweight footbridges in Flanders. The results indicate that the initial structural damping ratios for GFRP bridges exceed the values set in design guidelines and that they increase with an increasing pedestrian density. Further, it is shown that the measured accelerations do not relate proportionally to the pedestrian density. From both results the relevance of human–structure interaction is confirmed. Finally, while the first natural frequency is analytically predicted accurately, the vertical accelerations are substantially overestimated. Here, a better estimation can be made based on the experimentally measured damping ratios. The results contribute to a better understanding of human–structure interaction and the vibration assessment of lightweight footbridges. Practical applications include optimizing footbridge design, focussing on better performance and improving safety and user experience.

**Keywords:** lightweight footbridges; GFRP; vibration serviceability; comfort assessment; human–structure interaction

## 1. Introduction and Background

With the high demand from the community for more aesthetically pleasing and slimmer bridges in recent years [1–3], the influence of human–structure interaction on footbridges is becoming more important. In addition, improvements in materials, design methods, construction techniques and the involvement of architects has led to longer and slimmer bridges. As a result, current footbridges become more sensitive to human-induced vibrations [4–7], causing discomfort to pedestrians and endangering the use of the structure due to excessive vertical vibrations [8,9], even though the bridge is structurally safe to cross. Design codes address this dynamic problem by imposing natural frequency limits and vibration limitations to keep the footbridge experience enjoyable.

From a material point of view, the use of lightweight materials such as glass-fibre-reinforced polymers (GFRP) has increased significantly in bridge construction in recent decades [10–16]. In the Netherlands, thousands of pedestrian and road bridges for light and moderate traffic have already been constructed with this new and innovative material, and in the Flanders region of Belgium, the number of GFRP footbridges has increased

significantly by almost a factor of seven, from four small-scale projects in 2018 to twenty-seven footbridges with various spans and geometries in 2022 [17]. But outside the Benelux as well, more bridge construction projects are accomplished with GFRP materials, including projects in countries such as Sweden, Poland, Norway, UK, Australia, the United States of America, Dubai and China. Good mechanical properties, such as the high strength and stiffness-to-weight ratio, compared to traditional building materials, such as reinforced concrete and structural steel, play an important role [18,19]. In addition, the material has good durability, which is at least as important as the mechanical properties in making a choice, meaning less maintenance is required and that the bridge will have a low self-weight. Further, existing foundations can be reused, or less heavy lifting equipment is required for the installation, making this installation much faster [20–25]. However, due to this low structural self-weight, these types of footbridges will be more prone to dynamic problems that can influence the comfort for pedestrians on the bridge deck [26,27].

Human–structure interaction [28–37] plays an important role in these lightweight bridges, as several studies in the past have already shown. Indeed, the modal parameters, such as the first natural flexural frequency and the structural damping ratio, are significantly influenced [38–45] by it. Due to the interaction, the pedestrians, who both generate the vibration while walking and/or jogging on the bridge deck, as well as dampen this induced vibration by means of the ligaments in the human body [46], will strongly influence the comfort on the lightweight footbridges. However, current international design guidelines [47–50] do not take this human–structure interaction into account, which leads to a poor estimation of the comfort level and the structural damping ratio [51–53] during the design. The design formulas in these guidelines may lead to excessive material usage, especially for materials with a low overall stiffness such as GFRP, as the design of this type of footbridge is dominated by a serviceability limit state (SLS) (i.e., deflection, first natural flexural frequency and vibration comfort) [51,54]. Consequently, if a slender and aesthetic design is to be maintained, extra mass and therefore material will be necessary, leading to an excessive use of material to meet the requirements of the client, with an additional increase in production costs and environmental impact as a result.

Recent research by Gallegos-Calderón et al. [36,55], Ahmadi et al. [28,56] and Caprani et al. [57] on GFRP pultruded footbridges has shown that the results of acceleration responses under walking trials attained high accelerations despite meeting the 5 Hz design rule from Eurocode 0 [58]. The results also infer the use of interactive human models for a better representation of vibration responses from numerical models. In this paper, smartphone-based vibration measurements will be used to assess the modal parameters and vibration comfort on a selection of simply supported web-core sandwich footbridges. In contrast to measurements with dynamic sensors [59] (e.g., PCB 393B04 [60]), less investment is required and a large amount of data can be easily and quickly collected, analysed, processed and transmitted [61–64]. The first natural frequency range of this type of bridge is typically between 3 Hz and 15 Hz, and the accelerometer integrated in a smartphone in combination with a suitable application is a useful alternative. Research and comparison with professional dynamic sensors has also shown that the use of smartphones is suitable for dynamic structural monitoring [65,66]. The large amount of data also allows extensive analysis, comparison and control of the modal parameters of the different measurements.

This paper presents the implementation, execution and results of different vibration tests performed on four GFRP and four steel simply supported footbridges in the Flanders region of Belgium [67] during two measuring campaigns. The type and implementation of the vibration tests and the analysis of the vibration data are identical to those in [68]. As human–structure interaction is expected to occur in all lightweight footbridges, the dynamic behaviour of GFRP footbridges is compared with that of steel footbridges [69,70].

Following this introduction and background information, an overview is given of the various tested footbridges, including the location, the material, the abbreviations used in this paper and the geometric properties of the footbridges. The third part provides an overview of and the settings for the smartphone application iDynamics [71,72], followed

by an explanation of the applied test and measurement methods, i.e., implementation, locations of measurement points and number of pedestrians during tests, for the two measuring campaigns. Subsequently, the results of the first natural flexural frequencies and the structural damping ratios are presented for the two measurement sets. Based on the obtained results, conclusions and comparisons are made for the influence of human-induced vibrations and damping and human–structure interaction on the vibrational properties of lightweight footbridges. In the fourth part, the comfort of the lightweight GFRP and steel footbridges is examined based on the measured vertical accelerations during dynamic vibration tests with different amounts of walking and jogging pedestrians. In this part, the implementation method will first be discussed, after which the results and a comparison with analytical predictions are discussed. Finally, the main conclusions of this paper and the relevance of this study for the design of lightweight footbridges are summarized in the last part.

This paper will contribute to the understanding of human-induced vibrations and damping and further expand the knowledge related to human–structure interaction for lightweight GFRP and steel footbridges. The novelty of this paper in comparison with [68] lies in the reciprocal comparison of steel and GFRP footbridges and a new approach to better estimate the evolution of the structural damping value with increasing pedestrian density. Furthermore, a more accurate calculation based on experimentally obtained values of the structural damping at different pedestrian densities is implemented in the analytical calculation of the vertical accelerations in the design of lightweight footbridges. Practical applications include a better understanding of the observed range for the first natural flexural frequency and the structural damping ratio of short-span lightweight GFRP and steel footbridges, as well as the influence of the pedestrian density on both parameters. The results also point to shortcomings in the current standards and guidelines in connection to comfort and its analytical prediction. More in general, the findings may lead to optimizing footbridge design for better performance, considering human–structure interaction, assessing dynamic response and improving guidelines. The study emphasizes cost-effective design, improved safety and user experience. Overall, it contributes to creating sustainable and user-friendly footbridges.

## 2. Summary of the Examined Lightweight Footbridges

Table 1 gives an overview of the set (i.e., measurement campaign) number, the location in Flanders, the material of the main structure, the abbreviation used in the paper and the geometry (i.e., total length, width and surface area) of the eight lightweight footbridges that are studied in this paper. Besides this, a small description of the bridges and a picture can be found below the table and in Figure 1, respectively. As mentioned before, all bridges are simply supported and consist of a single span.

**Table 1.** Overview of the tested lightweight footbridges in the Flanders region.

| No. | Set | Year | Location | Material | Abbreviation | Length L (m) | Width W (m) | Surface Area A (m²) |
|-----|-----|------|----------|----------|--------------|--------------|-------------|---------------------|
| 1 | 1 | 2020 | Waregem | GFRP | GFRP_W | 10.00 | 4.00 | 40.00 |
| 2 | 1 | 2019 | Puurs | GFRP | GFRP_P | 16.60 | 4.20 | 69.72 |
| 3 | 1 | 2017 | Oudenaarde | Steel | Steel_O | 14.00 | 2.15 | 30.10 |
| 4 | 1 | 2017 | Sinaai | Steel | Steel_S | 17.50 | 3.30 | 57.75 |
| 5 | 2 | 2018 | Tremelo | GFRP | GFRP_T | 10.80 | 3.00 | 32.40 |
| 6 | 2 | 2012 | Tremelo | Steel | Steel_T | 33.20 | 3.00 | 99.60 |
| 7 | 2 | 2018 | Beersel | GFRP | GFRP_B | 7.00 | 2.00 | 14.00 |
| 8 | 2 | 2020 | Beersel | Steel | Steel_B | 10.00 | 2.20 | 22.00 |

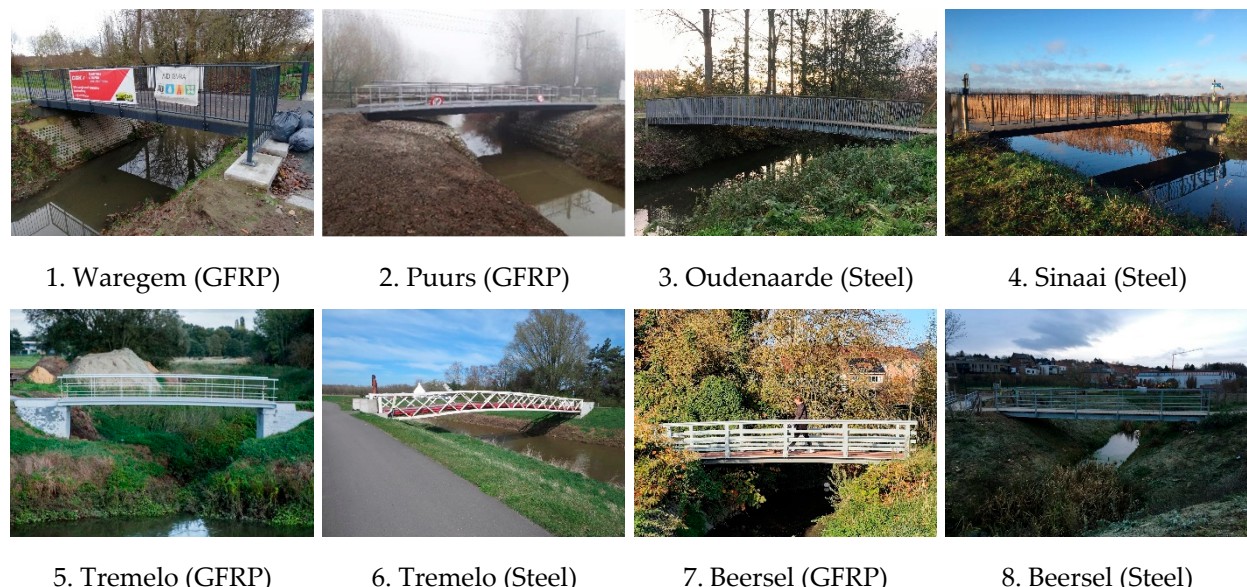

1. Waregem (GFRP)  2. Puurs (GFRP)  3. Oudenaarde (Steel)  4. Sinaai (Steel)

5. Tremelo (GFRP)  6. Tremelo (Steel)  7. Beersel (GFRP)  8. Beersel (Steel)

**Figure 1.** Studied bridges in the Flanders region of Belgium.

GFRP footbridge Waregem

Completed in 2020, this GFRP bridge over the Gaverbeek connects a parking lot with the neighbouring school.

GFRP footbridge Puurs

Along railway line 54 Mechelen—Terneuzen, located between Brussels and Antwerp, this GFRP bridge is part of Bicycle Highway F18, connecting the cities of Mechelen and Sint-Niklaas, and was installed in 2019.

Steel footbridge Oudenaarde

This bridge was realized in 2017 as part of the land development project 'Leie and Scheldt' by the Flemish government to replace the outdated previous bridge. The steel bridge spans the Zwalm river and provides an improved and safer connection for pedestrians and cyclists between the surrounding villages.

Steel footbridge Sinaai

In the autumn of 2017, the province of East Flanders, in collaboration with the municipality of Stekene and the city of Sint-Niklaas, completed this bridge over the Stekense Vaart next to an existing road bridge. It is the final piece of the infrastructure project Weimanstraat—Koebrugstraat, in which a safe bicycle connection was realized between Stekene and Sint-Niklaas.

GFRP footbridge Tremelo

Commissioned by the Flemish Waterway Department in 2018, this bridge, located at the confluence of the Laak and the Dijle, replaces an existing footbridge over the Laak.

Steel footbridge Tremelo

This steel girder bridge, taken into use in 2012, is also known as the Father Damien Bridge. It allows pedestrians and cyclists to cross the Dijle on the border of the municipalities of Haacht and Tremelo, close to the Tremelo GFRP bridge.

GFRP footbridge Beersel

Commissioned by Farys in 2018 at the intersection of the Broek and Dam in Beersel on the edge of the land development project 'Land van Teirlinck' [73,74], the bridge provides a connection between a school and the park behind it. Contrary to the other GFRP bridges in this study, the handrailings also entirely consist of composite material.

Steel footbridge Beersel

This bridge is one of three identical bridges that are part of the land development project 'Land van Teirlinck' commissioned by Flemish Land Company, the Municipality of Beersel and the Province of Flemish Brabant. The project aims to halt the increasing

risk of flooding of the Molenbeek valley in the village centres of Alsemberg and Sint-Genesius-Rode, in combination with facilitating accessibility for cyclists and pedestrians and increasing the touristic assets of the area.

## 3. Modal Testing, Vibration and Comfort Analysis Methods

Modal testing and parameter identification were performed using heel tests with different static pedestrian densities. This method was chosen since the response of the shorter structures to ambient excitation is too low to acquire good quality ambient vibration data [51]. During the heel test, one operator stands in the middle of the centre line of the bridge deck on the tips of the toes, after which their body weight falls on the heels. The impact load is repeated four to five times to detect anomalies and make a statistical interpretation. This heel test is a very simple test method that requires no additional test equipment (e.g., impact hammer) and produces accurate repetitive impacts on the bridge deck.

In the following sections, the iDynamics smartphone application and the measurement methods for the two measurement sets are described, after which the dynamic vibration tests and their elaboration are discussed.

### 3.1. iDynamics Based Smartphone Accelerometer Data Collection

iDynamics is a smartphone application for Android and IOS devices developed by TU Kaiserslautern, Germany. With the application, vibration measurements, system identification analyses (e.g., frequency and attenuation determination) and post-processing (e.g., filtering, smoothing and cutting of the vibration data) can be performed. Users are able to easily measure, process and export large amounts of vibration data using their smartphone. This makes it possible to carry out measurements on civil engineering structures with several devices simultaneously without the need for large investments in measuring equipment, and without losses in accuracy and repeatability. The internal accelerometer in the smartphone measures the acceleration along three main axes of the device.

During the tests, the smartphones are placed at the measurement points (MP) indicated in the next sections. In this paper, only the vertical vibrations perpendicular to the bridge deck plane are considered since lateral vibrations will only occur at higher frequencies due to the limited span, the considerable width and the tranverse stiffness of the tested bridges and are therefore difficult to generate by a pedestrian flow. With the smartphone placed on its back on the bridge deck, the vertical vibrations correspond to the Z-vibrations in the iDynamics application. For the collection of the vibrations, a measurement frequency (sample rate) of 50 Hz and a fast Fourier transformation (FFT) resolution of 512 are used. In general, and depending on the device and the internal accelerometer, accelerations between $0.0025 \text{ m/s}^2$ and $80 \text{ m/s}^2$ can be registered, widely covering the range of expected values.

For the analysis of the collected raw vibration data, all values are used in the post-processing, and the application calculates the maximum frequency weighted vibration strength $KB_{Fmax}$ directly from the vibration velocity history, according to the simplified method described in DIN 4150-2 [75,76]. In the calculation of the $KB_{Fmax}$ according to [76], the $c_F$ constant is by default set to 0.8, and this value is adopted in the post-processing of the vibration data. Before exporting the vibration data, a band-pass filter around the estimated first natural flexural frequency (i.e., +0.5 Hz and −0.5 Hz) of the bridge is applied to the raw vibration data to eliminate any second-order effects (e.g., resulting from the vibration of local deck elements).

### 3.2. First and Second Measurement Sets of Heel Tests

In this first measurement set, the pedestrians, with a random mass distribution, take their place uniformly along the length of the bridge and remain static during the execution of the heel test. The mass of the individual pedestrians was not recorded since a random distribution of the pedestrians is assumed during the different tests. However, the slight changes in the total weight of the pedestrians over the various tests on the different bridges

have a negligible effect compared to the overall influence of the pedestrians on the dynamic properties. Overall, an average weight of 70 kg per pedestrian is assumed. All measuring devices (i.e., smartphones) are placed at midspan at the location of the operator and register the evolution of the vertical accelerations as a result of the impact caused by the operator. Figure 2 shows the position of the operator (i.e., at MP1), the measurement points (MP) on the centreline and the distribution of the eight additional pedestrians.

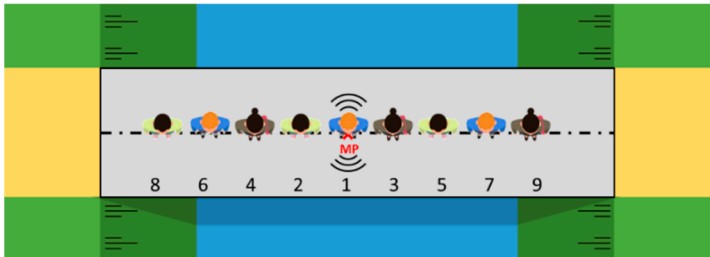

**Figure 2.** Test setup and distribution of pedestrians during the heel tests of set 1.

In contrast to the first measurement set, in the second set shown in Figure 3, the pedestrians, remaining static, stand in groups around/at a predetermined position on the bridge deck. In a first load case, abbreviated as LC1, the pedestrians are positioned in the middle of the bridge around the operator. In a second load case, abbreviated as LC2, the pedestrians stand on both sides of the bridge deck at L/4 from the supports. This pedestrian arrangement allows us to study the influence of the position of the pedestrians on the modal parameters. The results of the first measurement set should therefore lie between the results of LC1 and LC2 of the second measurement set.

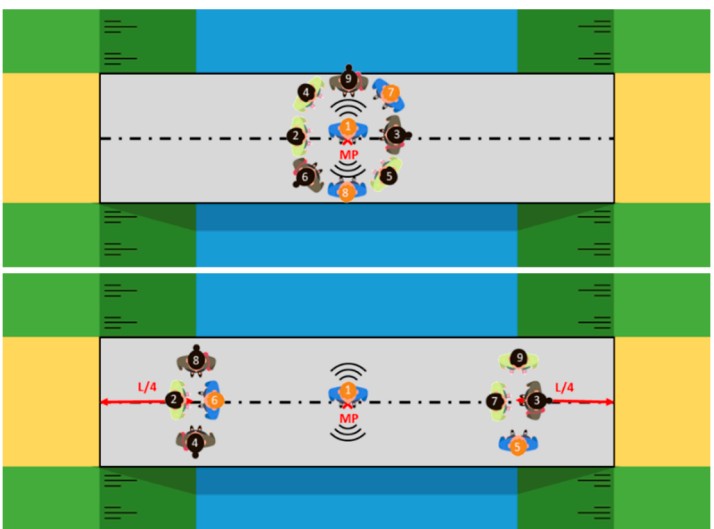

**Figure 3.** Test setup and distribution of pedestrians during load case LC1 (**top**) and LC2 (**bottom**) of set 2.

### 3.3. Modal Analysis

In this section, the processing of the modal parameters, i.e., the first natural flexural frequency and the structural damping ratio, will be discussed. Processing occurs after applying the bandpass filter [77–80], as discussed in Section 3.1, to the raw measured vibration data of the first and second measurement sets.

### 3.3.1. First Natural Flexural Frequency

As an example, Figure 4 displays the filtered vibration data in the three main directions of the first heel test with one pedestrian in the middle of the bridge deck on the GFRP

footbridge in Puurs (GFRP_P). As stated above, the vibrations in the horizontal X and Y directions are limited due to the centric application of the impacts and due to the bridge's high longitudinal and transverse stiffness. Hence, only the vibration in the z-direction will be considered, and the modal parameters will be determined for the vertical accelerations only. The higher harmonics of the natural flexural frequency are not considered as these are considered less relevant for this type of footbridge.

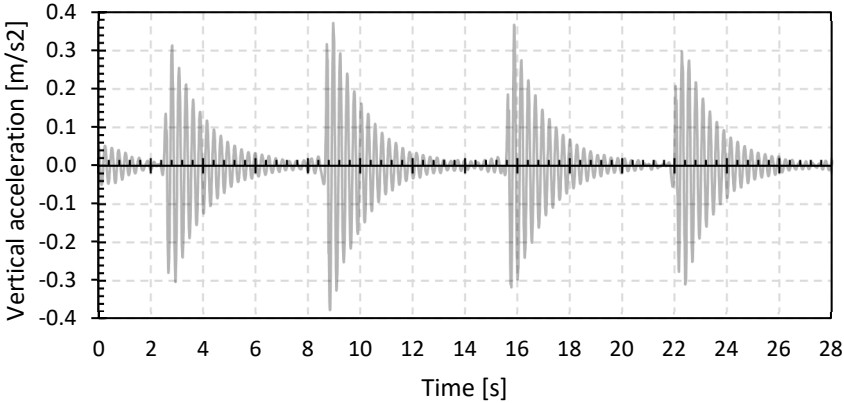

**Figure 4.** Filtered vibration data in the three main directions of a heel test with one pedestrian on the bridge deck for the GFRP footbridge in Puurs (GFRP_P).

Before processing the vibration data, a bandpass filter with low and high cut-off frequencies of, respectively, 0.5 Hz below and 0.5 Hz above the estimated first natural flexural frequency directly obtained from the iDynamics application is applied in the post-processing tab of the iDynamics application. This operation will remove second-order effects and outliers from the raw vibration data, resulting in data that are more suitable for the further determination of the first natural flexural frequency and the structural damping ratio [51,68].

After applying the filter to the raw vibration data, for every heel test, the processed z-vibration data are split into four individual vibrations (e.g., the second vibration in Figure 4 runs from 8 s to 14 s). The power spectrum density (PSD) graph is drawn from each vibration on the basis of a fast Fourier transform (FFT) with a minimum of 256 points. In the obtained PSD graph, the aliasing frequencies are removed, after which the first natural flexural frequency in the interval [0 Hz; 20 Hz] can be determined as the frequency having the largest PSD value. This method is repeated for all vibrations of the heel test, after which an average first natural flexural frequency of the vibrations can be determined.

### 3.3.2. Structural Damping Ratio

From the filtered z-vibrations of one heel test, the structural damping ratio along the positive (upward) and negative (downward) side of the vibration can be determined. First, the positive and negative peak values of the vibration must be determined. In this, a peak value is defined as an extreme positive or negative value of the vibration where the surrounding values are, respectively, smaller or larger than the respective value. A cut-off amplitude is used, which is equal to 5% of the maximum positive and negative amplitude of the individual vibration of a heel test. Values below this cut-off amplitude are considered signal noise and are not included in the determination of the peak values.

Based on the positive and negative peak values of the vertical acceleration of one individual vibration of a heel test and the equations below, the logarithmic decrement $\delta$ and the damping ratio $\zeta$ from the vibration can be determined. Here, the value of n will depend on the number of cycles that are considered.

$$\delta = \frac{1}{n} ln \frac{x(t)}{x(t+nT)} \tag{1}$$

$$\zeta = \frac{1}{\sqrt{1 + \left(\frac{2\pi}{\delta}\right)^2}} \tag{2}$$

Figure 5 shows the individual filtered z-vibration and the positive and negative peak values of the first vibration of the first heel test with one pedestrian (i.e., operator) in the middle of the bridge deck for the GFRP footbridge in Puurs (GFRP_P). From the inverse formula of the logarithmic decrement, a best-fit curve can be determined. The result of this best-fit positive and negative logarithmic decrement curve can also be found in Figure 5.

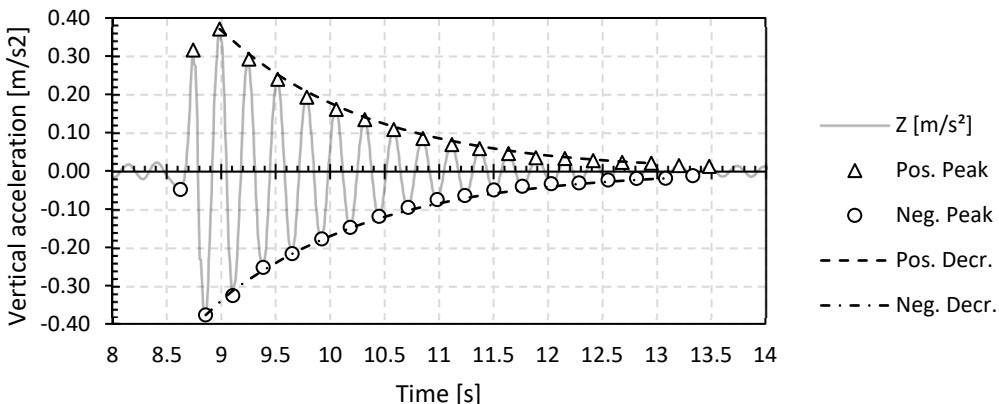

**Figure 5.** Logarithmic decrement of the second vibration based on the heel test with one person on the bridge deck for the GFRP footbridge in Puurs (GFRP_P).

The above procedure is repeated for all vibrations of a heel test, after which an average positive and negative value of the structural damping ratio is determined. The average of both the positive and negative structural damping ratio is considered the value of the structural damping for the relevant heel test measurement, pedestrian density (i.e., acronym $d_{TC}$) and footbridge.

### 3.4. Comfort Analysis of Lightweight GFRP and Steel Footbridges

In addition to the modal parameters of a footbridge, i.e., first natural flexural frequency and structural damping ratio, a comfort assessment based on the vertical accelerations is also executed. Previous research has shown that the pedestrian comfort on a footbridge is not only related to the first natural flexural frequency. To assess this comfort, dynamic vibration tests (DT) are performed on the aforementioned footbridges with various pedestrian densities. Contrary to the heel tests, the pedestrians on the bridge deck will no longer be static but will perform one of the two load situations, walking or jogging. The test setup is equal for sets 1 and 2.

Figure 6 provides a graphical representation of the test setup for the dynamic vibration tests. The measuring devices are placed along the centre line, at midspan and then at every 0.5 m in both directions. Depending on the bridge span and device availability, 6 to 11 measurement points are provided. The prescribed number of pedestrians, see Table 2, shall freely (i.e., without imposing an enforced walking or jogging frequency) walk or jog clockwise on the bridge deck for a period of two minutes.

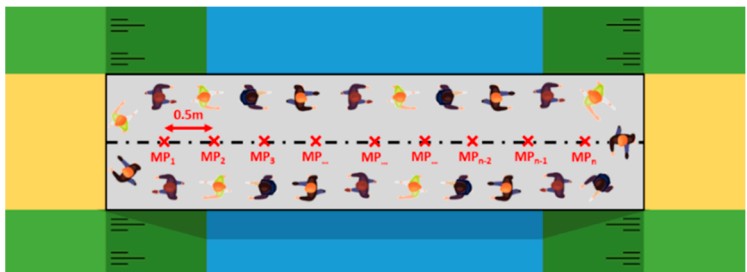

**Figure 6.** Test setup during the dynamic vibration tests on the footbridges (sets 1 and 2).

**Table 2.** Number of pedestrians and corresponding pedestrian densities during the dynamic vibration tests.

| No. | MC | Bridge | # Pedestrians (-) (Pedestrian Density (P/m²)) | | | | |
|-----|----|--------|------|------|------|------|------|
| | | | DT1 | DT2 | DT3 | DT4 | DT5 |
| 1 | 1 | GFRP_W | 4 (0.10) | 8 (0.20) | 20 (0.50) | 36 (0.90) | - |
| 2 | 1 | GFRP_P | 4 (0.06) | 7 (0.10) | 16 (0.23) | 24 (0.34) | - |
| 3 | 1 | Steel_O | 7 (0.23) | 14 (0.47) | 35 (1.16) | 42 (1.40) | - |
| 4 | 1 | Steel_S | 6 (0.10) | 12 (0.21) | 29 (0.50) | 42 (0.73) | - |
| 5 | 2 | GFRP_T | 4 (0.12) | 7 (0.22) | 10 (0.31) | 17 (0.52) | 33 (1.02) |
| 6 | 2 | Steel_T | 10 (0.10) | 15 (0.15) | 20 (0.20) | 25 (0.25) | 30 (0.30) |
| 7 | 2 | GFRP_B | 2 (0.14) | 3 (0.21) | 4 (0.29) | 7 (0.50) | 14 (1.00) |
| 8 | 2 | Steel_B | 3 (0.14) | 5 (0.23) | 7 (0.32) | 11 (0.50) | 22 (1.00) |

The prescribed number of pedestrians and the corresponding pedestrian density as a function of the bridge deck area (the value in brackets) can be found in Table 2. In the first measurement set, four different amounts of pedestrians were used, while in the second measurement set, five different dynamic vibration tests were performed.

In order to enable a comparison of the vertical acceleration and the associated comfort (criteria), a test criterion for the vertical accelerations must be established. For reference, the figures below were based on the GFRP footbridge in Puurs (GFRP_P). Figure 7 shows the absolute vertical z-accelerations for the measurements at MP8 for the load situation jogging with thirty-five pedestrians (pedestrian density 0.5 P/m²) for a time period of 100 s.

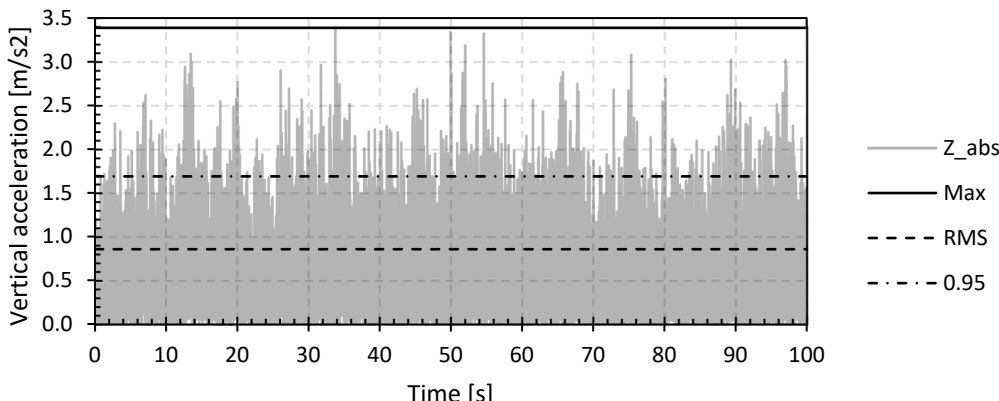

**Figure 7.** Absolute values of the vertical Z-accelerations and indication of the maximal, RMS and 95 percentile value for the GFRP footbridge in Puurs (GFRP_P) with 35 jogging pedestrians (0.50 P/m²).

Figure 7 also shows three possible assessment criteria for the vertical acceleration (a straight horizontal line over the time period), namely the maximum, root mean square (RMS) and the 95 percentile value of the absolute vertical accelerations. In international documents and guidelines, such as JRC-document EUR 23984 EN, HiVoSS RFS2-CT-2007-

00033 and SETRA [48,50,81], the choice is made to define the comfort of a structure based on the maximum occurring value of the vertical acceleration. However, in international literature, it is often opted to use the RMS value of the vertical accelerations [31,82–85]. Finally, the authors of this paper propose the 95 percentile value as an assessment criterion for the vertical accelerations.

Figure 7 indicates that by using the maximum absolute value of the vertical accelerations, an overestimation of the actual vibration behaviour and comfort of the bridge deck is likely to be obtained as in most cases this extreme acceleration will only occur for a very short period during walking or jogging on the bridge deck. If this absolute maximum value of the vertical accelerations is used as assessment criteria for the comfort of the footbridge, an unfavourable design of the footbridge with high material consumption will be obtained. The RMS value of the vertical accelerations on the other hand displays an underestimation of the actual comfort of the footbridge since accelerations higher than this RMS value will occur frequently. Basing the design on this value can therefore lead to discomfort during the service life. For the above reasons, the 95 percentile value of the vertical accelerations is chosen to estimate the vibration behaviour and comfort of the footbridge during the load cases of walking and jogging, which will be exceeded only by 5% of the time of the vibration. According to the authors, the 95 percentile value gives the best representation of the overall vertical accelerations, excluding any second-order effects which may arise, for example, from the deck planks.

## 4. Results of Parameter Identification of Lightweight GFRP and Steel Footbridges

The results of the modal parameters for the first and second measurement sets are presented in this section. In addition, the influence of the position of the pedestrians on the bridge deck based on the second measurement set is discussed. Finally, the results of the dynamic vibration tests on the different footbridges with different pedestrian densities and for the load situations walking and running are given. In all figures, the GFRP and steel bridges are indicated by black and white markers, respectively. The connections between the measured points, indicated by markers, are for clarity purposes only.

### 4.1. Heel Tests: First Measurement Set
4.1.1. First Natural Flexural Frequency

Figure 8 shows the results of the first natural flexural frequency as a function of the pedestrian density for the footbridges of the first set, i.e., with a uniform distribution of the pedestrians. The values shown are determined as the mean of six measurement results on the relevant bridge at the location of the measurement point in the middle of the bridge deck (as indicated in Figure 2). The deviation from the different measurement devices is limited. Indeed, the standard deviation per data sample point in Figure 8 varies between 0.9% and 2.4%.

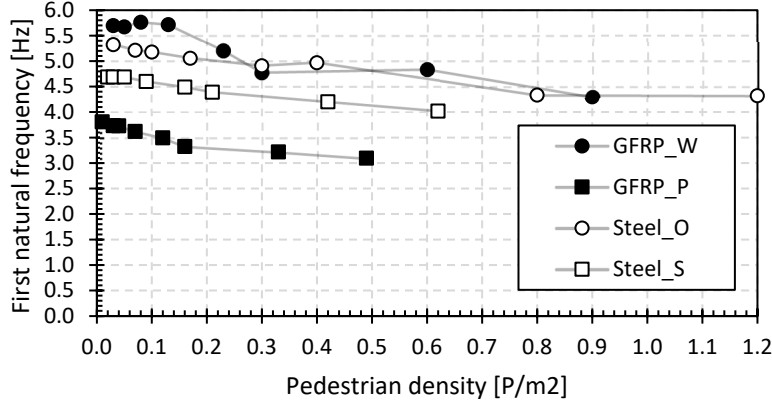

**Figure 8.** First natural flexural frequency for the footbridges of the first set.

Figure 9 shows the natural frequencies for different pedestrian densities, normalized by the natural frequency with only the operator on the bridge deck. This gives a representation of the relative change in natural frequency as a function of the pedestrian density.

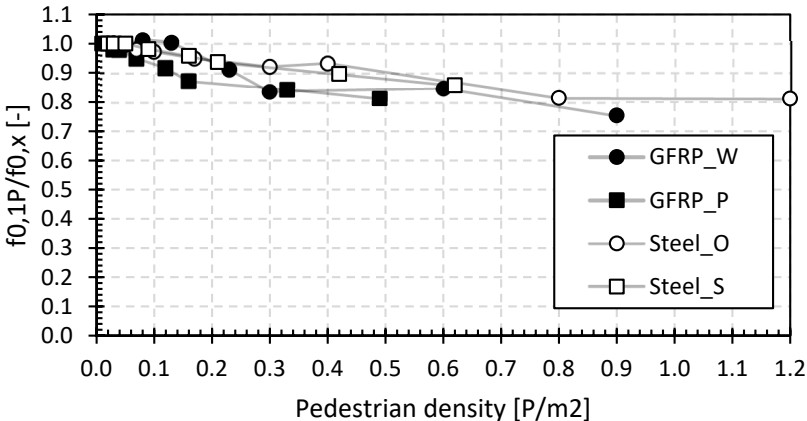

**Figure 9.** Relative first natural flexural frequency for the footbridges of the first set.

The measured basic frequencies range from 3.8 to 5.7 Hz, and these values reduce by up to 25% for higher pedestrian densities. These findings fall within the first natural frequency ranges reported in the research [51].

4.1.2. Structural Damping Ratio

Figure 10 shows the results of the structural damping ratio as a function of pedestrian density. The values were determined as the mean of six measurements at the middle of the bridge deck. The deviation from the different measurement devices is higher compared to the standard deviation per data sample in Figure 10 varying between 3.2% and 6.6%.

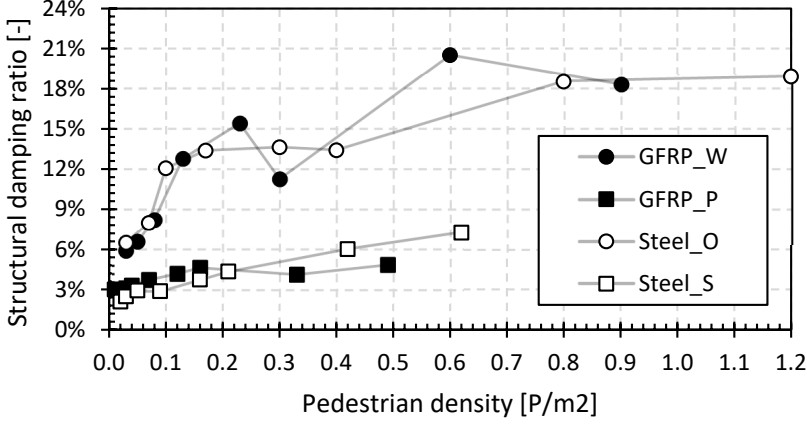

**Figure 10.** Structural damping ratio for the footbridges of the first set.

Figure 11 shows the damping ratios for different pedestrian densities normalized by the damping ratio with a single pedestrian load on the bridge deck in order to allow easy comparison in variation between the structural damping ratio values for the different pedestrian densities.

The structural damping ratios range from 2.1 to 6.5%, and these values increase to values from 4.8% to 20.5% for higher pedestrian densities. For all types of footbridges in this study (i.e., GFRP and steel), Figure 11 shows that the structural damping ratio will increase by a factor between 1.5 and 3.5 at higher pedestrian densities, proving the damping effect caused by the human body.

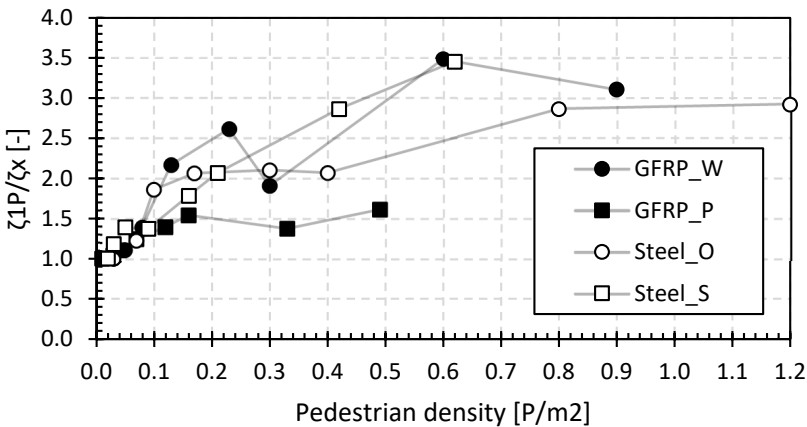

**Figure 11.** Relative structural damping ratio for the footbridges of the first set.

*4.2. Heel Test: Second Measurement Set*

4.2.1. First Natural Flexural Frequency

Figure 12 shows the results of the first natural flexural frequency as a function of the pedestrian density for the footbridges of the second set, for both load cases LC1 and LC2 (see Section 3.2). The values shown are determined as the mean of eleven (Tremelo) and eight (Beersel) measurement results on the relevant bridge at the location of the measurement point in the middle of the bridge deck (as indicated in Figure 2). The deviation from the different measurement devices is limited with the standard deviation per data sample point in Figure 12 varying between 1.2% and 1.3% for the footbridges in Tremelo and between 3.3% and 10.8% for the footbridges in Beersel.

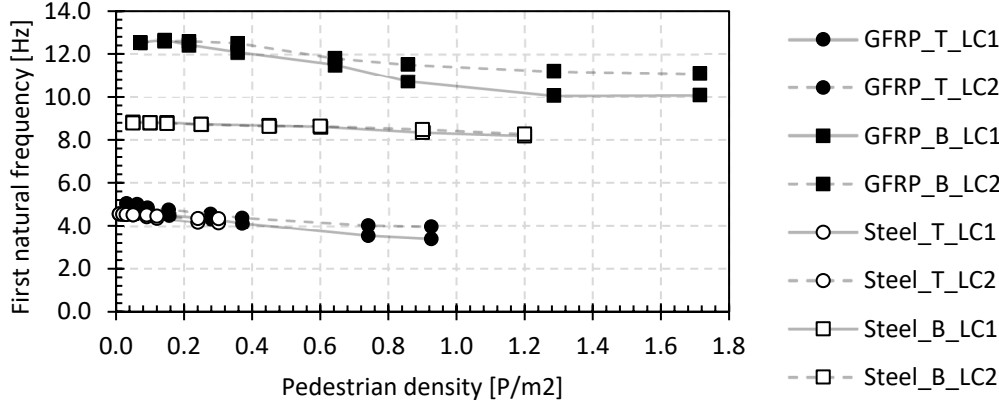

**Figure 12.** First natural flexural frequency of LC1 and LC2 for the footbridges of the second set.

The normalized natural frequencies between the results of LC1 and LC2 are shown in Figure 13. This ratio visualizes the influence of the position of the pedestrians, where a value of 1 indicates no influence of the position, while higher values demonstrate the higher modal mass of the pedestrians near the midspan. The position effect is noticeable for all bridges except for the steel bridge in Beersel (B_steel).

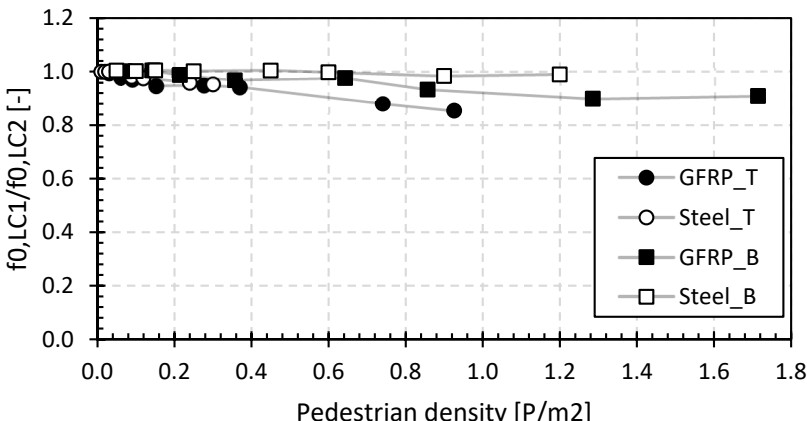

**Figure 13.** Relative ratio of LC1/LC2 for the first natural flexural frequency for the footbridges of set 2.

The measured basic frequencies range from 4.5 Hz to 12.5 Hz, and these values are reduced by up to 27% for higher pedestrian densities. The results again fall within the ranges reported in [51]. The relative relationship of the first natural flexural frequencies to that with only one pedestrian on the bridge deck is shown in Figure 14. It gives a representation of the relative change in natural frequency as a function of the pedestrian density. In this figure, the average values of the LC1 and LC2 measurements are used as the base for comparison.

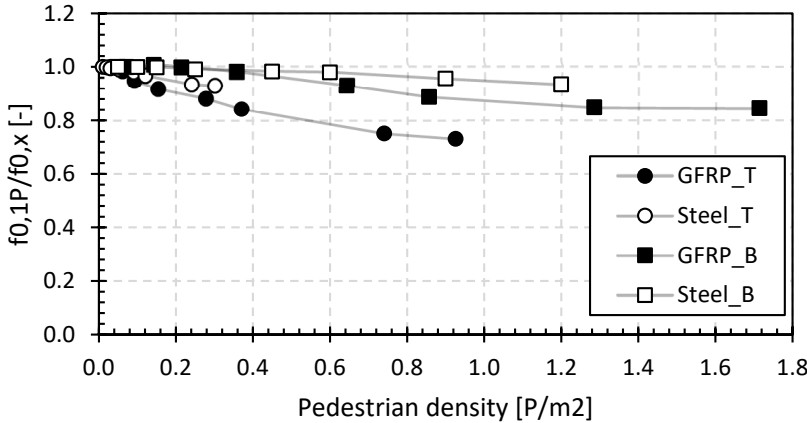

**Figure 14.** Relative average first natural flexural frequency for the footbridges of the second set.

### 4.2.2. Structural Damping Ratio

Figure 15 gives the mean values of the structural damping ratio as a function of the pedestrian densities for both load cases, LC1 and LC2. The relative relationship between the results of LC1 and LC2 is shown in Figure 16. This ratio visualizes the influence of the position of the pedestrians, where a value of 1 indicates no influence of the position, while higher values demonstrate the larger damping effect of the pedestrians near the midspan. Contrary to the position effect on the first fundamental frequency, here, the effect is most noticeable for the steel bridges while being small for the GFRP bridges. The average deviation between the measured values of the different measuring devices is between 10.5% and 15.6% for the footbridges in Tremelo and between 10.8% and 20.7% for the footbridges in Beersel. The deviation is larger than that for the first natural flexural frequencies in the second set in Figure 12, which is mainly due to the larger structural damping ratios at higher pedestrian densities, increasing the error during the calculation.

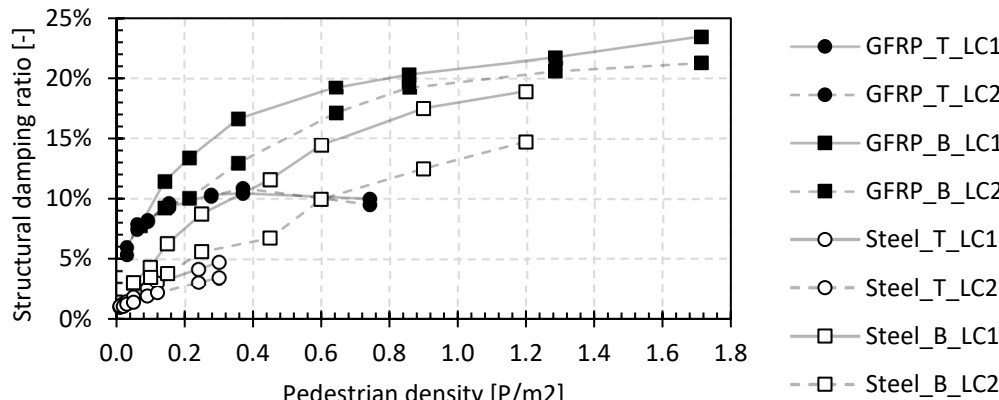

**Figure 15.** Structural damping ratio of LC1 and LC2 for the footbridges of the second set.

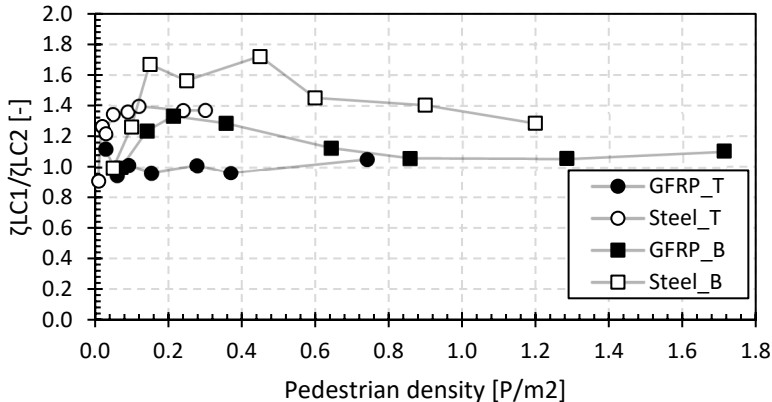

**Figure 16.** Relative ratio of LC1/LC2 for the structural damping ratio of the footbridges of the second set.

Figure 17 shows the relative ratio of the structural damping compared to a single pedestrian load averaged from results of the structural damping ratios of LC1 and LC2. The basic structural damping ratios range from 1.0% to 7.8%, and these values increase to values from 3.5% to 23.4% for higher pedestrian densities for all types of footbridges in this study (i.e., GFRP and steel). Figure 17 shows that the structural damping ratio will increase by a factor between 1.7 and 5.6 at higher pedestrian densities, proving the damping effect caused by the ligaments in the human body. Overall, the increase in the structural damping ratio for the GFRP and steel footbridges shows that there is human–structure interaction, which should not be underestimated.

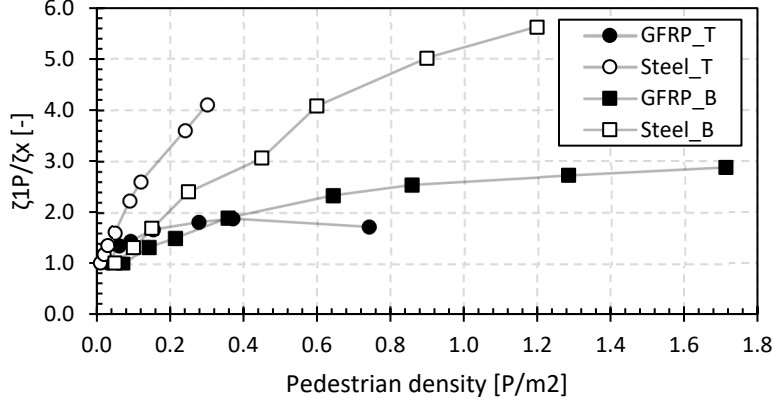

**Figure 17.** Relative structural damping ratio for the footbridges of the second set.

### 4.3. Dynamic Vibration Tests

Figure 18 summarizes the maximum, RMS and 95 percentile value of the vertical accelerations as a function of the pedestrian density for the two load situations, i.e., walking and jogging, for the GFRP footbridge in Puurs (GFRP_P) from the dynamic vibration tests (DT). The maximum, RMS and 95 percentile values are represented by triangular, square and circular markers, respectively, interconnected by a solid light grey line for jogging and a light grey dotted line for walking. The chosen assessment criterion of the 95 percentile value of the vertical accelerations is shown with a dark marker.

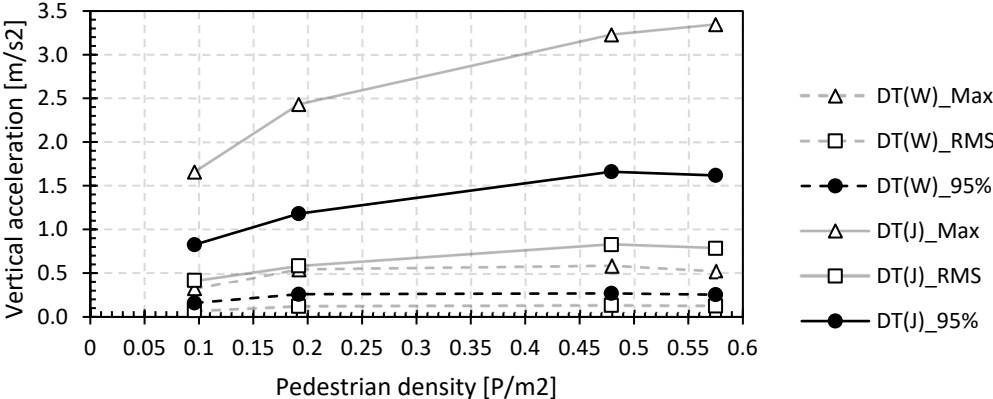

**Figure 18.** Maximum, RMS and 95 percentile values of vertical acceleration during walking and jogging on the GFRP footbridge in Puurs (GFRP_P) for different pedestrian densities.

From Figure 18, it can be seen that the maximum value of the vertical accelerations is approximately 2 times larger than the 95 percentile value, which on its turn is 1.8 times larger than the RMS value. The 95 percentile value forms a good middle ground between the maximum and RMS value, providing a good representation of the vibration behaviour and consequently the comfort of the footbridge. It should be noted that the effect of higher harmonics on the vertical accelerations of the lightweight footbridges under a walking or jogging load is not considered since in the vast majority of cases they will not be excited during a normal load situation.

#### 4.3.1. Comfort Assessment during Walking

Figure 19 shows the vertical accelerations as a function of the pedestrian density for all bridges and the load situation walking, determined as the mean of the 95 percentile over all measurement devices.

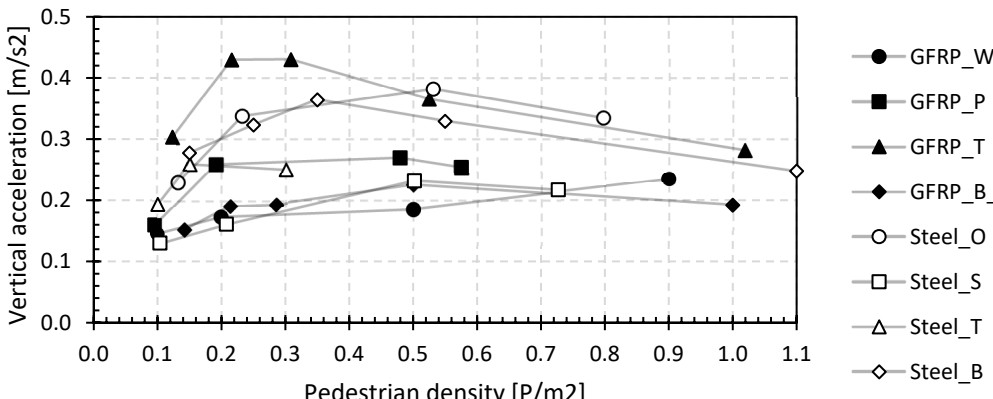

**Figure 19.** 95 percentile value of the vertical accelerations during walking.

The accelerations first increase with pedestrian density but then decrease again at higher pedestrian densities, except for the GFRP footbridge in Waregem, where there is a steady increase. For all types of footbridges, the accelerations are within the range of 0.13 to 0.43 m/s$^2$, which according to the JRC document for the design of lightweight footbridges [60] can be classified as maximum comfort (comfort class 1—CC1: $a_{vert.}$ < 0.5 m/s$^2$) [81].

Figure 20 shows the acceleration increase relative to the value with only the operator on the bridge as a function of the pedestrian density. Clearly, the accelerations do not increase proportionally with the densities, and from a certain density, a downward trend is, rather, observed. This is remarkable as a tenfold increase in pedestrian density only leads to a less than double acceleration. For the GFRP footbridge in Tremelo and the steel footbridge in Beersel, the vertical accelerations at high pedestrian densities are even below those at low densities. For the other bridges, the increase is limited between a factor of 1.3 and 2.1. It can therefore be said that a substantial amount of human–structure interaction is present in lightweight footbridges, irrespective of their material, which cannot be ignored in the design.

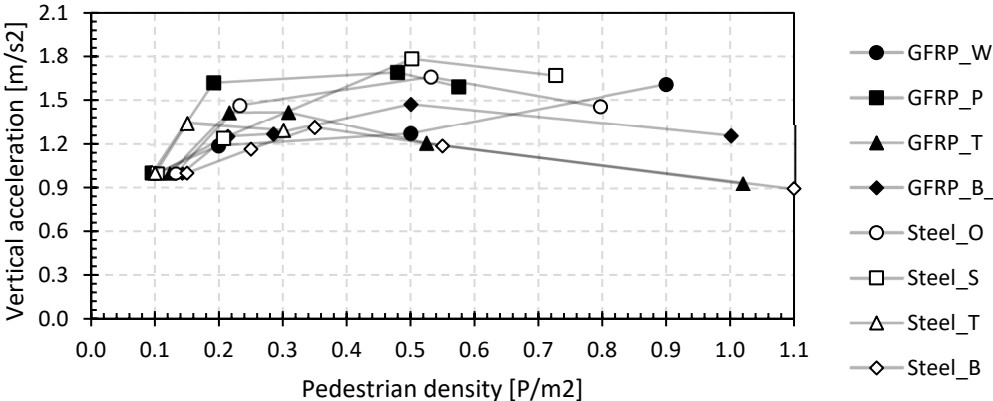

**Figure 20.** Relative vertical accelerations in function of the pedestrian density of the dynamic vibration test with the walking load case.

### 4.3.2. Comfort Assessment during Jogging

Figure 21 shows the vertical accelerations as a function of the pedestrian density for all bridges and the jogging load case. The vertical values were determined as the mean of the 95 percentile over all measurement devices.

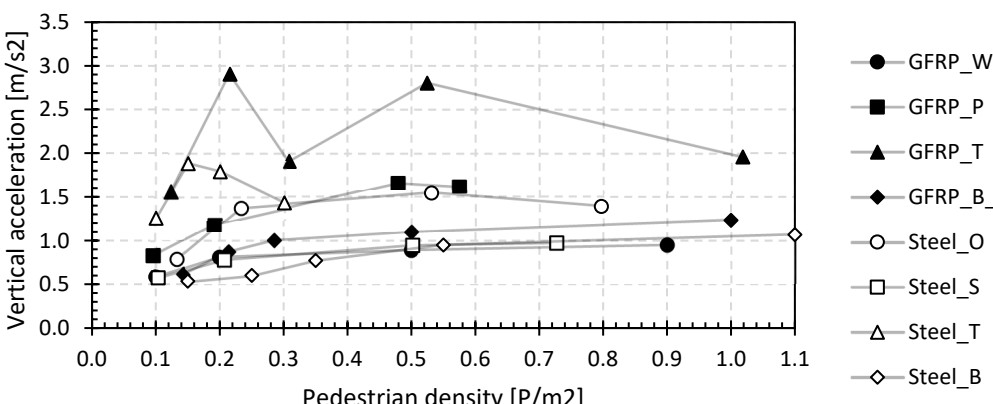

**Figure 21.** 95 percentile value of the vertical accelerations during jogging.

As for the walking load case, the vertical accelerations also increase with increasing pedestrian density, and for larger densities, the accelerations, rather, remain constant. The

values range between 0.54 and 2.90 m/s$^2$, corresponding from medium (0.5 m/s$^2$ > a$_{\text{vert.}}$ > 1.0 m/s$^2$) over minimal comfort (1.0 m/s$^2$ > a$_{\text{vert.}}$ > 2.5 m/s$^2$) to discomfort (a$_{\text{vert.}}$ > 2.5 m/s$^2$) according to the JRC document [81]. The relative acceleration increase as a function of the pedestrian density is shown in Figure 22.

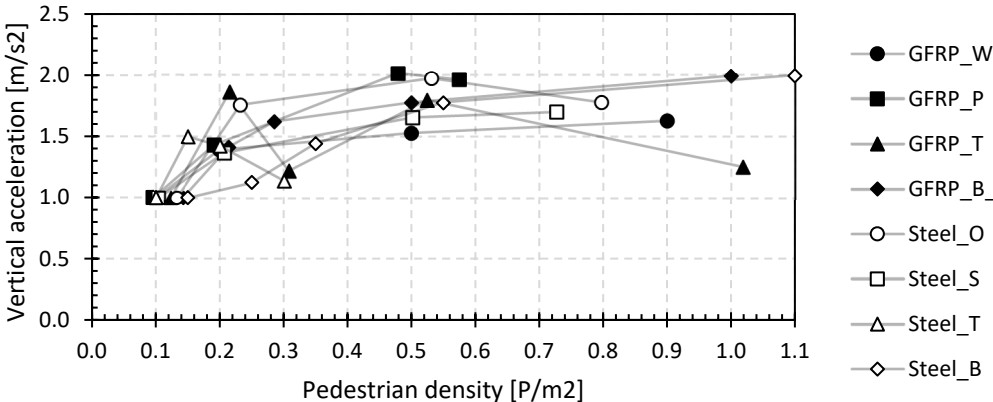

**Figure 22.** Relative vertical accelerations in function of the relative increase in pedestrian density of the dynamic vibration test with the jogging load case.

After an initial stronger increase at low densities, the increase is rather limited thereafter. The increase lies in the intervals 1.1 and 2.0. Also, for jogging, the accelerations will not increase proportionally. Pedestrian discomfort on small bridges during jogging, although not included in most specifications, is remarkable. Only medium or minimum comfort, even considering 95 percentile values, is reached with only a few joggers on the bridge, while such discomfort is never reached during walking, even at high densities of 1.0 P/m$^2$.

## 5. Discussion of the Modal Parameters and Comfort Assessment

### 5.1. General Observations

Figures 23 and 24, respectively, show the first natural flexural frequency and the initial structural damping ratio (only the operator on the bridge) for the tested bridges with spans from 7.0 m to 33.20 m. The simply supported GFRP footbridges in Maldegem, Ghent, Lille and Galmaarden from previous research [68] are also added to both figures. As already mentioned in previous sections, the first natural frequency for the GFRP and steel footbridges is in the range of 3.80 Hz to 12.5 Hz, and for the structural damping ratio, it is in the range of 0.95% and 7.75%. These values coincide well with the observations made in [51].

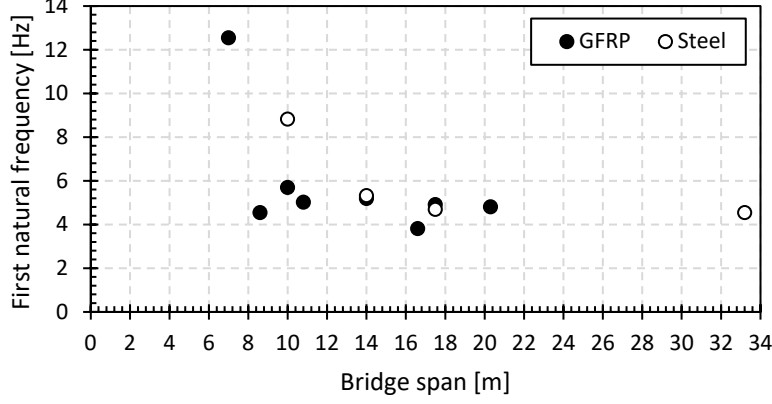

**Figure 23.** First natural flexural frequency in function of the bridge span.

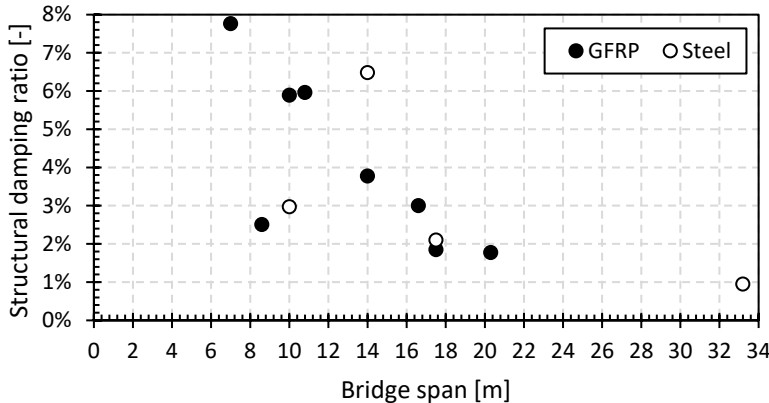

**Figure 24.** Initial structural damping ratio in function of the bridge span.

The first natural frequency decreases by 2% to 20% at 0.5 P/m$^2$ with increasing pedestrian density. The decrease mainly depends on the structural mass and thus the dimensions of the bridge in question. These findings are in accordance with the formulas for the first natural flexural frequency of [86] as shown in Figure 25, where the addition of non-structural mass by the pedestrians on the bridge causes a reduction in the natural flexural frequency. The magnitude of the reduction in the first natural flexural frequency depends on the ratio between the total non-structural mass of the pedestrians to the mass of the bridge, where the total mass of the pedestrians is limited by the usable surface area of the bridge deck.

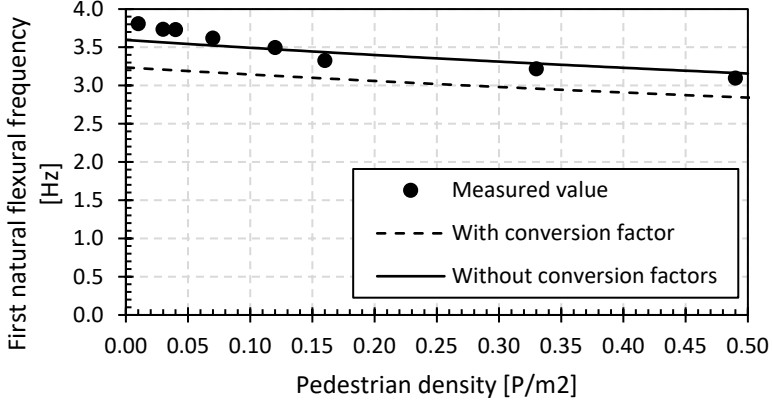

**Figure 25.** Evolution of the first natural flexural frequency for the GFRP footbridge in Puurs (GFRP_P): measured vs. calculated.

The initial structural damping ratios are higher for the tested GFRP (3% to 8%) than for the steel footbridges (1% to 6.5%). Especially for the GFRP bridges, they are considerably higher than those stated in international guidelines at 0.5% to 1% [81,87]. For all tested footbridges, the structural damping ratio increases significantly with the pedestrian density. The increase varies between a factor of 1.5 and 5.5 at a pedestrian density of 0.5 P/m$^2$. The effect of the location of the pedestrians in the second set is detectable but limited [51]. This localized dynamic loading causes localized areas of higher structural damping as the bridge dissipates more energy via the pedestrians on the bridge, which in turn absorb this energy into the body ligaments to accommodate the increased dynamic response. The magnitude of the absorbed energy, and consequently the increase in the structural damping ratio of the bridge, is strongly dependent on the number and positioning (static or dynamic) of the pedestrians on the bridge deck.

## 5.2. Relation between Modal Parameters and Vertical Accelerations

From Figures 8, 12, 19 and 21, it can be concluded that a high natural flexural frequency (i.e., $f_0 > 5$ Hz [58]) does not necessarily lead to an acceptable comfort level, certainly for the load case of jogging. For example, the GFRP footbridge at Puurs (P_GFRP) and the steel footbridge at Sinaai (S_Steel) have an initial value of the first natural frequency (only the operator on the bridge deck) of, respectively, 3.81 Hz and 4.69 Hz, which are lower than 5 Hz, but they will not have the largest vertical accelerations under the load cases of walking and jogging. It can be concluded that the guidelines specified in Eurocode 0 do not apply to the comfort analysis of lightweight footbridges and that, in absence of better guidelines, additional in situ vibration analyses should be performed on this type of bridge.

By combining Figures 11 and 17 of the relative structural damping ratios with Figures 20 and 22 giving the relative vertical accelerations, the influence of the structural damping on the vertical accelerations can be discussed. It can be concluded that the increasing structural damping ratio caused by human–structure interaction [86,88,89] will significantly reduce the vertical accelerations, especially at higher pedestrian densities. The pedestrians will on the one hand cause the vibrations on the bridge deck but will on the other hand also act as a damper due to the ligaments in the human body. As more pedestrians are present on the bridge deck, the effect of the human damping will gain the upper hand, limiting the vertical accelerations of the bridge deck during walking and jogging.

## 6. Analytical Comparison

In this section, a brief analytical comparison for the GFRP footbridge in Puurs (P_GFRP) is given. The reader is referred to [68] for the theoretical background and the properties of the bridge.

### 6.1. First Natural Flexural Frequency

The first natural flexural frequency values at different pedestrian densities, as measured by the heel tests for the GFRP footbridge in Puurs (see Figure 8), are given in Figure 25. In addition to the measured values, the analytically calculated values are given, based on both initial and reduced mechanical properties [68]. These reduced mechanical properties take into account material (i.e., geometric deviations, model uncertainties and uncertainties in the properties of the material) and conversion factors for the effects of environmental factors (i.e., moisture and temperature) and the aging (i.e., creep and fatigue) of the GFRP material.

Given the recent installation of the GFRP footbridge in Puurs in 2019, the measured values of the first natural flexural frequency should correspond to the non-converted (initial) values. From Figure 25, it can be concluded that there is indeed a very good agreement between the measured and calculated values for the different pedestrian densities with a mean standard deviation of only 11%.

### 6.2. Comfort Analysis

The comfort of the bridge can be analysed based on the vertical accelerations of the bridge deck at a certain pedestrian density. In what follows, the analytical prediction is carried out, following the JRC document [81] and the procedure explained in [68] based on the Dutch guideline CUR96:2019 [87]. Contrary to the comparison made in [68] by the authors, the updated and density-dependent measured damping ratios can now be used in the analytical prediction formula. Here, in the determination of the analytical vertical acceleration, the linearly interpolated density-dependent damping ratios from Figure 26, based on the experimentally measured values of Figure 10, are used.

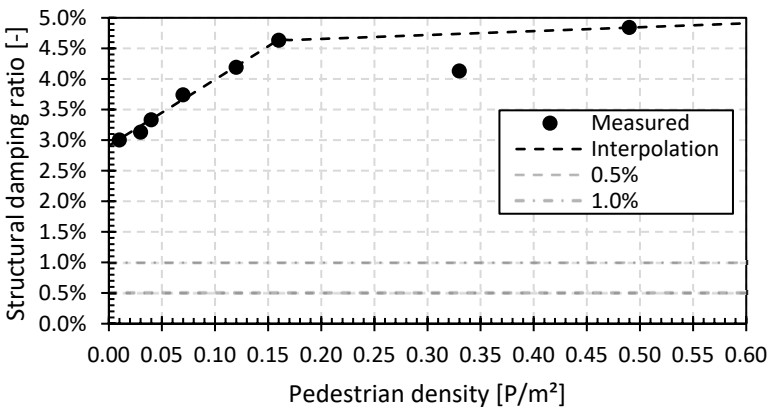

**Figure 26.** Evolution and linear interpolation of the structural damping ratio for the GFRP footbridge in Puurs (GFRP_P).

The results of the calculation for the design vertical accelerations (i.e., indicated in this paper by $a_{d,vert}$) calculated according to [81] for different pedestrian densities and different structural damping ratios (i.e., 0.5% and 1.0% according to CUR96:2019 and the structural damping ratio according to the interpolation in Figure 26) are presented in Figure 27. In addition to the calculated values, the measured maximum, RMS and 95 percentile values of the vertical accelerations for four pedestrian densities for the load case of walking for the GFRP footbridge in Puurs (P_GFRP) are shown with the triangular, square and circular marker, respectively.

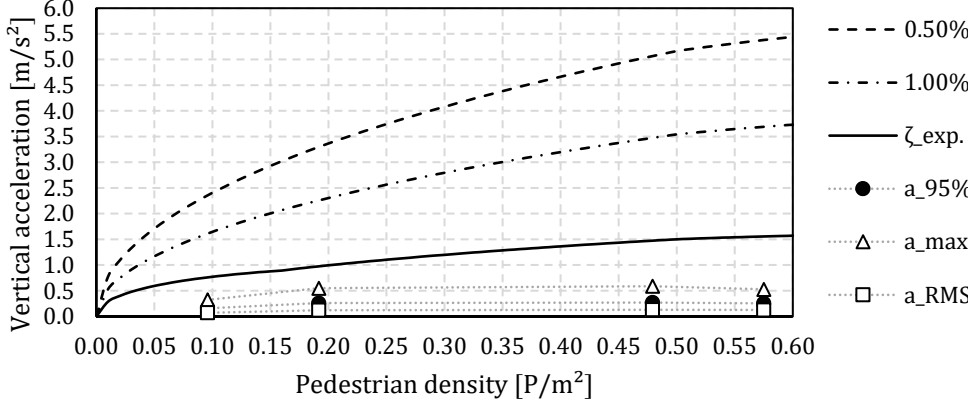

**Figure 27.** Comparison of measured and predicted accelerations for the GFRP footbridge in Puurs (GFRP_P).

From Figure 27 it can be deduced that on the basis of the minimum and average values of the structural damping ratio according to CUR96:2019 (i.e., 0.5 an 1%) only a maximum comfort, linked to comfort class 1 (i.e., $a_{d,vert}$ < 0.5 m/s$^2$) according to [81], can be obtained for very small pedestrian densities (i.e., $d_{TC}$ < 0.01 P/m$^2$). Indeed, the vertical accelerations will rise quickly and soon reach comfort class 4 with discomfort at a pedestrian density of 0.11 P/m$^2$ ($\approx$1P) and 0.24 P/m$^2$ ($\approx$2P) for 0.5% and 1.0%, respectively.

If, however, measured structural damping ratios (including human–structure interaction) are used, comfort class 2 will be reached at a pedestrian density of 0.04 P/m$^2$ until 0.21 P/m$^2$, after which it will be limited to maximum comfort class 3 (i.e., 1.0 m/s$^2$ < $a_{d,vert}$ < 2.5 m/s$^2$) with minimum comfort. These predictions are closer to the actual measured maximum vertical acceleration values. It can be argued that better agreement between analytically calculated and measured maximum vertical accelerations can be found for the walking load case if experimentally obtained values of the structural damping ratio that take into account the human–structure interaction are used, although the predictions still

overestimate the actual vibration levels. For the jogging load case, there are currently no guidelines. In view of the observations made in Section 4.3.2, this is a lack in the regulations.

## 7. Conclusions

In this paper, a study is presented on the modal parameter identification and comfort assessment of GFRP and steel lightweight footbridges in relation to human–structure interaction. For this, heel and dynamic vibration tests were used. From this study, the following main conclusions can be drawn:

- The basic value of the first natural flexural frequencies for the GFRP footbridges is between 3.8 Hz and 12.5 Hz, and for the steel footbridges, it is between 4.5 Hz and 8.8 Hz. The first natural frequency will decrease by 2% to 20% with increasing pedestrian densities up to 0.5 $P/m^2$. The initial values of the structural damping ratio with only the operator on the bridge deck are between 3.0% and 8.0% for the GFRP footbridges and between 1.0% and 6.5% for the steel footbridges. The structural damping ratio will increase by a factor between 1.5 and 5.5 with increasing pedestrian density at 0.5 $P/m^2$. In addition, the position of the pedestrians on the bridge deck has a small but noticeable influence on the increase in the structural damping ratio. The position of the pedestrians on the bridge deck influences the dynamic loading pattern and, consequently, the bridge's response. For example, a large group of pedestrians walking in unison or concentrated in a specific area can cause localized dynamic loading and excite specific modes of vibration in the bridge. This can lead to localized areas of higher structural damping. Furthermore, pedestrians positioned near structural members or sensitive regions of the bridge may cause higher dynamic responses in those areas, affecting the overall damping behaviour of the structure. These observations point at significant human–structure interaction for small lightweight footbridges.
- The vertical accelerations are acceptable for the walking load case (CC1) but quickly become unacceptable for the jogging load case (CC3 and CC4). In the international guidelines, only the accelerations for walking are considered. As a result of the increasing damping, the experimentally measured 95 percentile vertical acceleration values do not increase proportionally with increasing pedestrian density, contradicting international guidelines. Excluding human–structure interaction therefore leads to uneconomic designs of lightweight footbridges. Despite the high basic first natural flexural frequency ($f_0$ > 5 Hz) of some of the tested lightweight GFRP and steel footbridges, the comfort will be minimal (CK3) or even worse for the jogging load case. The statement of Eurocode 0, declaring that no check of the vibrations should be carried out if the first natural flexural frequency is larger than 5 Hz, is therefore not valid for these lightweight footbridges. It is recommended to always carry out an in situ check of the vibration behaviour and comfort.
- Good agreement between calculated and measured first natural flexural frequency values for the GFRP bridge in Puurs can be obtained by the analytical formula from the Dutch guideline CUR96:2019. Following the current guidelines in combination with the recommended damping ratios (0.5% and 1.0%) for GFRP bridges stated in CUR96:2019 clearly overestimates the vertical accelerations for the walking load case. However, if experimentally obtained pedestrian density-dependent structural damping ratio values are used, better agreement can be obtained between the analytically predicted and the experimentally measured maximum acceleration values for the walking load case.

**Author Contributions:** Conceptualization, W.D.C. and W.V.P.; methodology, W.D.C. and W.V.P.; software, J.U.; validation, J.U.; formal analysis, J.U.; investigation, J.U.; resources, W.D.C.; data curation, J.U.; writing—original draft preparation, J.U.; writing—review and editing, J.U., W.D.C. and W.V.P.; visualization, J.U. and W.D.C.; supervision, W.D.C. and W.V.P.; project administration, W.D.C.; funding acquisition, W.D.C. All authors have read and agreed to the published version of the manuscript.

**Funding:** This research was funded by Research Fund Flanders (FWO) grant number 1S50522N.

**Data Availability Statement:** The raw/processed data required to reproduce these findings cannot be shared at this time as the data also form part of an ongoing study.

**Conflicts of Interest:** The authors declare no conflict of interest.

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
