# Peer review of "Modal Parameter Identification and Comfort Assessment of GFRP Lightweight Footbridges in Relation to Human–Structure Interaction"

_jcs, doi:10.3390/jcs7090348_

Round 1

Reviewer 1 Report

The reviewer comments can be found in file attached.

Regards

A Moderate editing of English language required.

Reviewer 2 Report

I would like to commend the authors on their manuscript. I found it interesting and very well written. I have some observations:

P1          Line 41: “and endangering the use of the structure” – please clarify endangering.

P3          Line 94: “predictions aree discussed”: are?

P3          Line 104: respectively – not required.

P3          Table 1: is the span the length of the unsupported section?

P5          Line 164: “the raw collected vibration data” – the collected raw vibration data

P5          Explain KBFmax and why a value of 0.8. How is “strength” calculated?

P5 and onwards: were the masses of the participants measured and did these affect the results, and their distribution?

P7          Figure 4 – it is very difficult to discern between x, y and z due to the shade of grey.

P8          Line 240: “and at every 0.5 m from thereon” – “and thence at every 0.5 m in both” …

P10        Line 269: “an unfavourable design of the footbridge will be obtained” – define unfavourable.

P16        Line 387: define CC1

P16        Line 388, 396: “relative acceleration increase as function of”: as a function of

P16        Line 396: “relative increase in pedestrian density” – is this relative?

P16        Line 388: “shows the relative acceleration increase as function of the pedestrian density” – relative to what – define the relative increase from which value – 0, 0.1 P?

P17        Line 405: avert – vert is subscript?

P19        Line 455: “reduced properties” – how were these calculated? Vapour (line 457) do you mean moisture absorption into the matrix?

P19        Line 470: “For this, the simplified linear interpolation from Figure 26, based on Figure 10, is used.” Please explain.

P20        Line 473: “ad,vert” explain.

P20        various comfort classes are introduced for the first time, without an introduction nor explanation.

P21        Conclusions: there are far too many. They should be grouped into different main aspects to highlight the principal design recommendations for GFRP bridges for the interested reader. 

English is of a high standard.

Round 2

Reviewer 1 Report

Dear Authors,

Since the last reviewer's comments are not applied completely in the revised manuscript and just some minor changes have been done carelessly, I invite the authors to make a major revision and reconsider the manuscript again.

Regards

 Moderate editing of English language required

Reviewer 2 Report

I would commend the authors - all queries have been addressed.